# Some *t*-tests for *N*-of-1 trials with serial correlation

**Jillian Tang**¤, **Reid D. Landes**\*

Department of Biostatistics, University of Arkansas for Medical Sciences, Little Rock, Arkansas, United States of America

¤ Current address: Department of Computer Science, Stanford University, Stanford, California, United States of America

\* rdlandes@uams.edu

## Abstract

*N*-of-1 trials allow inference between two treatments given to a single individual. Most often, clinical investigators analyze an individual's *N*-of-1 trial data with usual *t*-tests or simple non-parametric methods. These simple methods do not account for serial correlation in repeated observations coming from the individual. Existing methods accounting for serial correlation require simulation, multiple *N*-of-1 trials, or both. Here, we develop *t*-tests that account for serial correlation in a single individual. The development includes effect size and precision calculations, both of which are useful for study planning. We then use Monte Carlo simulation to evaluate statistical properties of these serial *t*-tests, namely, Type I and II errors, and confidence interval widths, and compare these statistical properties to those of analogous usual *t*-test. The serial *t*-tests clearly outperform the usual *t*-tests commonly used in reporting *N*-of-1 results. Examples from *N*-of-1 clinical trials in fibromyalgia patients and from a behavioral health setting exhibit how accounting for serial correlation can change inferences. These *t*-tests are easily implemented and more appropriate than simple methods commonly used; however, caution is needed when analyzing only a few observations.

**Data Availability Statement:** All software code and data used in this paper are available at https://github.com/rdlandes/red-face; see files starting with "N-of-1". These files can also be provided upon request.

## 1. Introduction

Behavioral scientists have used *N*-of-1 trials for more than a century [1]. Guyatt *et al.* (1986) brought these trials, with randomized crossover designs, to the attention of mainstream medical research in the 1980s (see also Gabler *et al.*, 2011) [2, 3]. In this era of personalized medicine, *N*-of-1 trials are appearing in medical research with increasing frequency [4]. Consequently, guidelines for these studies were added to the Consolidated Standards of Reporting Trials (CONSORT) in 2015; see CONSORT Extension for *N*-of-1 Trials (CENT) [5]. CENT guidelines call for statistical methods that account for within-subject correlation. This call echoes what reviews of *N*-of-1 studies often note: *N*-of-1 data exhibit serial correlation (e.g. first-order autocorrelation), and most studies fail to account for serial correlation. [1, 3, 6, 7].

This work develops a formula-based statistical method for *N*-of-1 studies that accounts for serial correlation while using only the data from a single individual to draw inferences. Most existing methods emerged with increases in computing power. These methods typically

**Funding:** The authors received no specific funding for this work.

**Competing interests:** The authors have declared that no competing interests exist.

provide inference on two types of differences between two treatments: level- and rate-change. Level-change is when the difference in means is not dependent on the time series of the treatments, whereas rate-change is when the difference in means *is* dependent on the time series of the treatments. Rochon (1990) describes a large-sample, maximum likelihood method that evaluates both level- and rate-change, but no closed-form estimator exists [8]. Hence, an iterative procedure produces the estimates. McKnight *et al.* (2000) developed a double-bootstrap method for making inference on level- and rate-change [9]. Their first bootstrap estimates serial correlation; the second uses the estimated correlation to compare two treatments. They provide statistical properties for their method, and they focus on trials having as few as 20 or 30 observations. Borckardt and company describe statistical properties of the Simulation Modelling Analysis for *N*-of-1 trials, and consider trials having between 16 and 28 observations from an individual [10, 11]. Simulation Modelling Analysis is similar to a parametric bootstrap method, with the bootstrap method generating replicates under the null hypothesis. Empirical *p*-values for level- and rate-change result. Lin *et al.* (2016) propose semiparametric and parametric bootstrap methods (only one bootstrap needed) for evaluating level- and rate-change [12]. They explore the statistical properties of their method for trials having 28 observations. Other *N*-of-1 methods exist, but the methods described here are the only ones we could find that use only the observations from a single individual and account for serial correlation.

All of the methods above are computationally intensive and require either special software or substantial statistical expertise. However, researchers conducting *N*-of-1 trials seem to prefer simpler analysis methods. Gabler *et al.* (2011) reviewed analyses conducted in 108 *N*-of-1 trials and found 52% used visual analysis, 44% used *t*-tests, and 24% used nonparametric methods (some studies used more than one analysis method) [3]. Punja *et al.* (2016) reviewed 100 reports of conducted (60%) and planned (40%) *N*-of-1 trials [13]. Seventy-five of these performed or planned statistical analyses: 53% of these 75 used paired *t*-tests and 32% used a nonparametric method. Though several of these simple analysis methods use only the observations from one individual, they fail to account for serial correlation. A substantial proportion of researchers using *N*-of-1 trials sacrifice their need for appropriate analyses to their desire for simplicity. Our goal in this work is to tend to their analytical needs and desires by developing a simple method that uses only the data from a single individual.

Furthermore, researchers and clinicians typically collect a small number of observations from an individual in an *N*-of-1 trial. In the Gabler *et al.* (2011) review, the median number of outcomes measured on an individual was 20 with a range of 3–512. They also found the median number of crossovers was 3, with a range of 1 to 11 [3]. Similarly, Punja *et al.* (2016) reported a median of 3 repeated cycles / treatment blocks, with the range of repeated cycles from 2–15 and treatment blocks from 2–5 [13]. For this reason, we focus primarily on realistically sized *N*-of-1 trials; that is, *N*-of-1 trials with 12 or less crossovers, or 24 observations or fewer observations in total.

We develop four *t*-tests for individual differences between two treatments. While using only the observations from a single individual, these tests accommodate serial correlation (Section 2); we refer to these as "serial *t*-tests." Two of these serial *t*-tests evaluate level-change, and the other two evaluate rate-change. Within the two level- and the two rate-change tests, one test assumes observations from two treatments are paired in a series over time; we call these "paired serial *t*-tests." The other test assumes one treatment's observations are independent of the other's; we call these "2-sample serial *t*-tests." In Section 3, we use simulation to compare Type I error rates and estimated power between the four serial *t*-tests and their "usual *t*-test" analogues that ignore serial correlation. Then, we compare the serial *t*-tests' estimated power to their theoretical power. In Section 4, we illustrate how these tests may be used in clinical-trial and behavioral-health settings. We discuss the results in the final section.

## 2. Serial *t*-tests

We first describe *N*-of-1 trial or study designs to which our proposed serial *t*-tests apply. Next, we give the general development of the serial *t*-test, followed by paired and 2-sample serial *t*-tests for level-change, then paired and 2-sample serial *t*-tests for rate-change. Sample size considerations for study planning follow. Finally, since the serial *t*-tests require an estimated correlation, we recommend a serial correlation estimator.

### 2.1. Applicable *N*-of-1 designs

The CONSORT extension for reporting *N*-of-1 trials (CENT) lists several single-case designs that have been referred to as *N*-of-1 trials, but CENT currently only considers multiple withdrawals/reversals, "ABAB," or multiple-crossovers designs as "*N*-of-1" in its scope (see Fig 1 in [14]) [5, 14]. For the purpose of analysis using the serial *t*-tests developed below, our definition of *N*-of-1 trial designs is broader than that of CENT. These serial *t*-tests may be used in the following *N*-of-1 study designs to compare two treatments, say A and B, using data from only one individual. Given that minimum size requirements are met (defined below for each test), the 2-sample serial *t*-tests may be applied to *N*-of-1 trials when there is one withdrawal/reversal ("AB" or bi-phase), multiple baseline, or changing criterion designs (see Fig 1 in [15]). The 2-sample serial *t*-tests may also be applicable to *N*-of-1 studies with pre-post designs. The paired serial *t*-tests may be used to analyze *N*-of-1 randomized trials with multiple withdrawals/reversals or crossovers, sometimes called alternating treatment designs [15]; i.e., those deemed true *N*-of-1 trials by CENT guidelines.

### 2.2. The serial *t*-statistic

The general model for a single series of *m* observations coming from an individual is $Y \sim N(X\boldsymbol{\beta}, R\sigma^2)$, where *R* is a first-order autoregressive correlation matrix, with element (*j,k*) defined as $\rho^{|j-k|}$ for *j,k* = 1,2,. . .,*m*. We refer to $\rho$ as the serial correlation. For the two-sample models described below, we assume additional series of observations coming from the individual have the same $\sigma^2$ and $\rho$.

Constructing the *t*-statistic requires estimators of $\boldsymbol{\beta}$ and $\sigma^2$, but not $\rho$ (see Section 2.5). We use ordinary least squares (OLS) estimators because, in the end, they provide formula-based test statistics that may be computed by those having little statistical expertise. (Generalized least squares estimation gives the best linear unbiased estimator of $\boldsymbol{\beta}$ and an unbiased estimator of $\sigma^2$, but requires an iterative process.) The OLS estimator of $\boldsymbol{\beta}$ is $\hat{\boldsymbol{\beta}} = (X'X)^{-1}X'Y$ for *X* of full column rank, and is unbiased. Under the assumed model, $Y \sim N(X\boldsymbol{\beta}, R\sigma^2)$, $\hat{\boldsymbol{\beta}}$ has variance $(X'X)^{-1}X'RX(X'X)^{-1}\sigma^2$ for a single series of *m* observations. For the design matrices, *X*, which we define in Sections 2.3 and 2.4, $\text{Var}(\hat{\boldsymbol{\beta}})$ is a function of *m* and $\rho$; we write

$$\text{Var}(\hat{\boldsymbol{\beta}}) = c(\rho) \times \sigma^2$$

suppressing dependence on *m* since it is known.

Turning attention to the variance, $\sigma^2$, the OLS estimator is $s^2 = \{Y'(I-P_X)Y\}/\{m-\text{rank}(P_X)\}$. However, $s^2$ is biased when $\rho \neq 0$; $E(s^2) = [\{m-\text{tr}(P_X R)\}/\{m-\text{rank}(P_X)\}]\sigma^2$ with $\text{tr}(P_X R)$ not equal to the rank of the projection matrix, $P_X = X(X'X)^{-1}X'$. For the design matrices defined below, the bias is a function of *m* and $\rho$; we thus re-write

$$E(s^2) = b(\rho) \times \sigma^2 \tag{1}$$

suppressing dependence on known *m*. Additionally, we may express $b(\rho)$ in terms of the

effective sample size of the series, $m'$, and the number of estimated location parameters, $p$:

$$b(\rho) = \frac{m(m' - p)}{m'(m - p)} \qquad (2)$$

[16]. Note, $m'$ depends on $\rho$; we omit that dependence though to reduce notation burden. Following from Eqs 1 and 2, an unbiased estimator of $\sigma^2$ may be written as

$$\tilde{s}^2 = s^2 \Big/ b(\rho) \qquad (3A)$$

$$= \frac{m'(m - p)}{m} s^2 \Big/ (m' - p) \qquad (3B)$$

For the serial *t*-statistic testing whether scalar $\beta = \beta_0$, the standard normal random variable is $(\hat{\beta} - \beta_0)/\sqrt{c(\rho) \times \sigma^2}$. (Without loss of generality, we henceforth let $\beta_0 = 0$.) The $\chi^2$ divided by its degrees of freedom is easily identified in Eq 3B and concisely written in Eq 3A. The serial *t*-statistic is then

$$t_{DF} = \frac{\hat{\beta}}{\sqrt{c(\rho) \times s^2 \Big/ b(\rho)}} \qquad (4)$$

with DF = $m' - p$. Thus, $\hat{\beta} \pm t_{DF,\alpha/2}\sqrt{c(\rho) \times s^2/b(\rho)}$ provides a $(1-\alpha)100\%$ confidence interval for $\beta$.

## 2.3. Paired and 2-sample serial *t*-tests for level-change

For the paired serial *t*-test for level-change, observations from treatments A and B come from a series of pairs (Fig 1A). We take the difference in observations from A and B in the same pair as our *Y* (Fig 1B); since this test is for level-change, we assume the mean of differences, $\mu_{A-B}$, does not depend on the series. The mean model is $X\boldsymbol{\beta} = \mathbf{1}_m[\mu_{A-B}]$, with $\mu_{A-B}$ estimated by the sample mean, $\overline{Y}_{A-B}$.

The $c(\rho)$, $b(\rho)$, and *m'* for this and the following 2-sample level-change test are

$$c_L(\rho) = \frac{m + 2\rho^{m+1} - m\rho^2 - 2\rho}{m^2(\rho - 1)^2} \qquad (5)$$

$$b_L(\rho) = \frac{m(1 - c_L(\rho))}{m - 1} \qquad (6)$$

$$m_L' = \frac{m}{m - (m - 1)b_L(\rho)} \qquad (7)$$

where the *L* subscript denotes level-change. The paired serial *t*-statistic for level-change is then

$$t_{DF} = \frac{\overline{Y}_{A-B}}{\sqrt{c_L(\rho) \times s^2 \Big/ b_L(\rho)}} \qquad (8)$$

with $DF = m_L' - 1$. Replacing $\rho$ with its estimate, $r$ (see Section 2.5), the statistic becomes

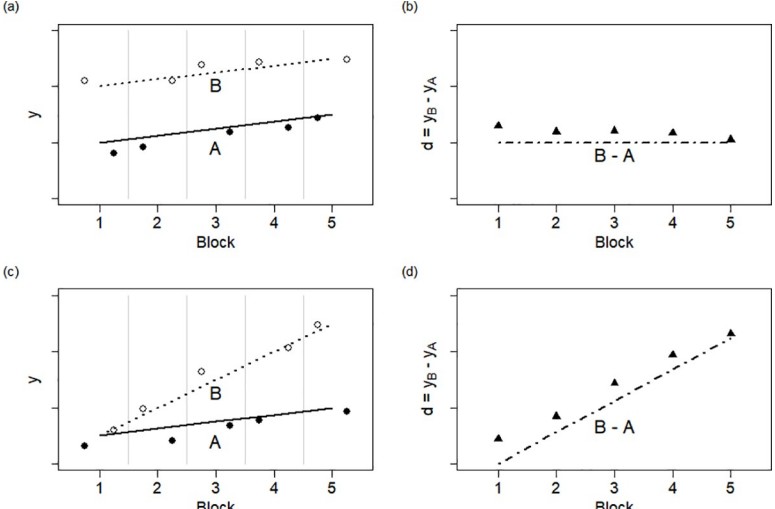

**Fig 1.** These simulated data represent *N*-of-1 trials with *m* crossovers of treatments A and B, randomized within block (left panels). The differences between A and B within a block (right panels) may be suitably analyzed with paired serial *t*-tests for level-change (top panels) and for rate-change (bottom panels). The true means are represented with lines, and serially correlated observations with points.

approximately $t_{DF}$. We note that the minimum *m* for this test is 4; this is because three parameters are estimated: $\mu_{A-B}$, $\sigma^2$, and $\rho$.

For the 2-sample serial *t*-test for level-change, observations come from a series particular to the treatment, and the two series may be treated as independent (Fig 2A). As the test is for level-change, we assume the difference between means, $\mu_A - \mu_B$, does not depend on the series. We treat the original two series of observations, $Y_A$ and $Y_B$, as independent, so

$$X\boldsymbol{\beta} = \begin{bmatrix} \mathbf{1}_{m_A} & \mathbf{0}_{m_A} \\ \mathbf{0}_{m_B} & \mathbf{1}_{m_B} \end{bmatrix} \begin{bmatrix} \mu_A \\ \mu_B \end{bmatrix}$$

As with the paired test, $\mu_A$ and $\mu_B$ are estimated by the sample means, $\overline{Y}_A$ and $\overline{Y}_B$. The *t*-statistic is

$$t_{DF} = \frac{\overline{Y}_A - \overline{Y}_B}{\sqrt{\left(\frac{c_{L_A}(\rho)}{b_{L_A}(\rho)} + \frac{c_{L_B}(\rho)}{b_{L_B}(\rho)}\right) \times s^2}} \tag{9}$$

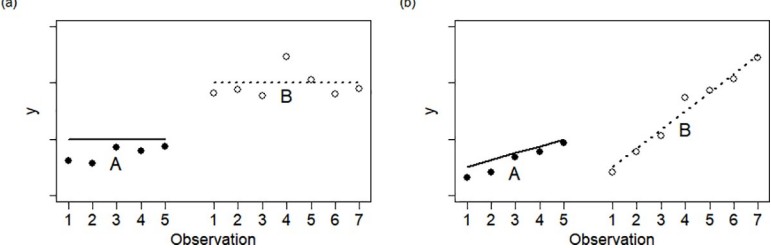

**Fig 2.** These simulated data represent *N*-of-1 trials where a series of observations from treatment A are observed first, followed by a series of observations from treatment B. These data may be suitably analyzed with 2-sample serial *t*-tests for level-change (a) and for rate-change (b). The true means are represented with lines, and serially correlated observations with points.

with $DF = m'_{L_A} + m'_{L_B} - 2$. Replacing $\rho$ with a pooled estimate, $\bar{r} = (m_A r_A + m_B r_B)/(m_A + m_B)$, the statistic becomes approximately $t_{DF}$. For this test, the minimum $m$ requirements are both $m_A$ and $m_B$ are 3 or more, and $m_A + m_B \geq 7$, since we estimate the overall $\sigma^2$ as well as $\rho$ and $\mu$ for each treatment.

## 2.4. Paired and 2-sample serial *t*-tests for rate-change

For the paired serial *t*-test for rate-change, observations from two treatments again come from a series of pairs (Fig 1C). We take the difference in observations from A and B in the same pair as our *Y* (Fig 1D); however, since this test is for rate-change, we assume the mean of $\mathbf{Y}_{A-B}$ depends linearly on the series. Then

$$X\boldsymbol{\beta} = \begin{bmatrix} \mathbf{1}_m & \mathbf{x} \end{bmatrix} \begin{bmatrix} \mu_{A-B} \\ \beta_{A-B} \end{bmatrix}$$

where we use $\mathbf{x}$ centered about 0; i.e., $\mathbf{x}' = (1,2,\ldots,m)-(m+1)/2$. The parameter of interest is $\beta_{A-B}$; it is estimated by the slope on $x$ from a simple linear regression.

For the serial paired *t*-test for rate-change test and the following 2-sample serial *t*-test for rate-change, $c(\rho)$, $b(\rho)$, and $m'$ are defined as follows:

$$c_R(\rho) = \frac{12}{(m^2-1)^2} \left( \begin{array}{c} -\dfrac{6\rho(\rho+1)^2(\rho^m-1)}{m^2(\rho-1)^4} \\[2mm] +\dfrac{2\rho(6\rho^{m+1}+6\rho^m+\rho^2-2\rho+1)}{m(\rho-1)^3} \\[2mm] -\dfrac{6\rho(\rho^m+1)}{(\rho-1)^2} - \dfrac{2m\rho}{\rho-1} + \dfrac{m^2-1}{m} \end{array} \right) \tag{10}$$

$$b_R(\rho) = \frac{1}{m-2}\left( m-1 - \frac{2\rho(\rho^m-m\rho+m-1)}{m(\rho-1)^2} - \frac{m(m^2-1)c_R(\rho)}{12} \right) \tag{11}$$

$$m'_R = \frac{2m}{m-(m-2)b_R(\rho)}, \tag{12}$$

with the *R* subscript denoting rate-change. The *t*-statistic for the paired serial *t*-test for rate-change is then

$$t_{DF} = \frac{\hat{\beta}_{A-B}}{\sqrt{c_R(\rho) \times s^2 / b_R(\rho)}} \tag{13}$$

where $DF = m_R' - 2$ and $\rho$ is replaced with its estimate, $r$, to get an approximate $t_{DF}$. This test requires $m \geq 5$ as four parameters are estimated.

For the 2-sample serial *t*-test for rate-change, observations come from a series particular to the treatment, and the two series may be treated as independent (Fig 2B). Since the test is for rate-change, we assume the means of $Y_A$ and $Y_B$ depend linearly on the series. We treat the original

two series of observations, $Y_A$ and $Y_B$, as independent series, so the mean model of $Y = [Y_A, Y_B]'$ is

$$X\boldsymbol{\beta} = \begin{bmatrix} X_{m_A} & \mathbf{0}_{m_A} \\ \mathbf{0}_{m_B} & X_{m_B} \end{bmatrix} \begin{bmatrix} \mu_A \\ \beta_A \\ \mu_B \\ \beta_B \end{bmatrix} \tag{14}$$

where $X_{m_*} = \begin{bmatrix} 1_{m_*} & x_{m_*} \end{bmatrix}$. $\beta_A$ and $\beta_B$ are estimated with the slopes from an ordinary linear regression having the $X\boldsymbol{\beta}$ in Eq 14. The *t*-statistic is

$$t_{DF} = \frac{\hat{\beta}_A - \hat{\beta}_B}{\sqrt{s^2 \left( \frac{c_{R_A}(\rho)}{b_{R_A}(\rho)} + \frac{c_{R_B}(\rho)}{b_{R_B}(\rho)} \right)}} \tag{15}$$

with $DF = m'_{R_A} + m'_{R_B} - 4$ and replacing $\rho$ with its pooled estimate, $\bar{r} = (m_A r_A + m_B r_B)/(m_A + m_B)$ as before, for the approximate $t_{DF}$. Here, both $m_A$ and $m_B$ must be 4 or more and $m_A + m_B \geq 9$ in order to estimate the seven parameters.

## 2.5. Sample size considerations when planning *N*-of-1 trials

Desired precision of estimated effects and/or power dictate the number of observations (*m*) needed from an individual undergoing an *N*-of-1 trial. Other than Rochon (1990), all of the authors reviewed in Section 1 considered power for their proposed methods, but only for specified sample sizes and/or effect sizes [8–12]. Wang & Schork (2019) followed up on Rochon's work, investigating power, but only considered trials with *m* = 400 [17]. None of the aforementioned authors gave guidance on how to compute sample sizes. Senn (2017) presented methods for computing sample sizes in studies involving *N*-of-1 trials, given desired power or precision [18]. Unfortunately, Senn's methods are for computing how many *trials* are needed, which assumes multiple *N*-of-1 trials are analyzed together; additionally, his methods did not account for serial correlation. Percha et al. (2019) developed tools to inform experiment design and estimate power by simulating *N*-of-1 trials under various controlled conditions and random effect assumptions; however, their tools do not account for serial correlation [19].

Our serial *t*-tests allow precision, sample size, effect size, and power calculations with existing routines or functions in standard statistical software (e.g., R, SAS). The assumptions for these calculations are the same as for usual *t*-tests, but also require an assumed serial correlation, $\rho$.

Both precision and effects size calculations require values for the significance level, sample size (*m*), and serial correlation ($\rho$). Substituting the values of *m* and $\rho$ into the formulae for $c(\rho)$, $b(\rho)$, and $m'$ from the desired serial *t*-test, precision, expressed as an expected margin of error for a $(1-\alpha)100\%$ confidence interval for $\beta$, is $t_{DF,\alpha/2} \sqrt{c(\rho) \times \sigma^2 / b(\rho)}$, where $\sigma^2$ is often replaced by 1, but may also be replaced by an assumed value. For computing an effect size, $\delta$ (in $\sigma$ units), given a specified power, the following steps outline how to use an existing effect size calculator for a one-sample *t*-test.

- First note the calculator's noncentrality parameter, $\delta_{(C)} \sqrt{m_{(C)}}$, and degrees of freedom, $DF_{(C)} = m_{(C)} - 1$, where subscript (*C*) denotes the calculator's parameters.

- Using the assumed value of $\rho$, compute DF (a function of $m'$), and $c(\rho)$ for the desired serial *t*-test.

**Table 1. Expected margins of error for 90% confidence intervals (95% one-sided confidence limit) for $\mu_{A-B}$ from the paired serial *t*-test for level change in Eq 8; assumes $\sigma^2 = 1$.**

| $\rho$ | $m = 4$ | 5 | 6 | 7 | 8 | 9 | 10 | 11 | 12 |
|---|---|---|---|---|---|---|---|---|---|
| 0 | 1.18 | 0.95 | 0.82 | 0.73 | 0.67 | 0.62 | 0.58 | 0.55 | 0.52 |
| 0.2 | 1.81 | 1.37 | 1.14 | 0.99 | 0.89 | 0.82 | 0.76 | 0.71 | 0.67 |
| 0.4 | 3.61 | 2.38 | 1.83 | 1.52 | 1.31 | 1.17 | 1.07 | 0.99 | 0.92 |
| 0.6 | 14.78 | 7.00 | 4.43 | 3.24 | 2.58 | 2.16 | 1.88 | 1.67 | 1.52 |
| 0.8 | 1272.65 | 214.23 | 70.60 | 33.06 | 19.06 | 12.55 | 9.05 | 6.96 | 5.61 |

- Set $m_{(C)}$ equal to DF+1. Usually, $m_{(C)}$ will not be a whole number; the calculator must be able to accept positive real numbers for the degrees of freedom parameter (or sample size, $m_{(C)}$).

- Using $m_{(C)}$, compute $\delta_{(C)}$ with the calculator.

- Setting the noncentrality parameter for the desired serial *t*-test equal to the calculator's non-centrality parameter, the effect size is $\delta = \delta_{(C)}\sqrt{m_{(C)}c(\rho)}$.

To assist in planning an *N*-of-1 trial for which the paired serial *t*-test for level change (Eq 8) is the intended analysis. Table 1 gives expected margins of error for 90% confidence intervals, and Table 2 gives $\mu_{A-B}$ values (in $\sigma$ units) detectable with 0.80 power on a one-sided 0.05 significance level test. In the supplementary material, we provide R code for generating these tables for all 4 serial *t*-tests allowing users to specify different confidence coefficients and power/Type I error values.

## 2.6. Estimating a serial correlation

The serial *t*-tests require an estimate of serial correlation, $\rho$. For a $\rho$ estimator, we work with the residuals, $e$, of the data from their estimated means:

$e_j = y_{A-B,j} - \overline{y}_{A-B}$ for the paired serial *t* for level-change;

$e_{ij} = y_{ij} - \overline{y}_i$ for the 2-sample serial *t* for level-change;

$e_j = y_{A-B,j} - \hat{\mu}_{A-B} - \hat{\beta}_{A-B}x_j$ for the paired serial *t* for rate-change;

$e_{ij} = y_{ij} - \hat{\mu}_i - \hat{\beta}_i x_{ij}$ for the 2-sample serial *t* for rate-change, where $i = A,B$ and $j = 1,2,\ldots,m_i$. Note $\overline{e} = 0$; this fact simplifies the maximum likelihood estimator of of $\rho$ to $\hat{\rho} = \sum_{j=2}^{m} e_j \times e_{j-1}/ \sum_{j=1}^{m} e_j^2$. However, $\hat{\rho}$ has bias of $-2\rho/(m-1)+O(\rho m^{-2})$ [20]. For small sample sizes, the bias is substantial. We sought another estimator of $\rho$.

Solanas *et al.* (2010) evaluated performance of ten serial correlation estimators [21]. Importantly, they considered sample sizes between 5 and 20. Among the ten estimators, they found

**Table 2. Effect sizes of $\mu_{A-B}$ (in $\sigma$ units) detectable with 0.80 power on a 0.10 (one-sided 0.) level test, using the paired serial *t*-test for level change in Eq 8.**

| $\rho$ | $m = 4$ | 5 | 6 | 7 | 8 | 9 | 10 | 11 | 12 |
|---|---|---|---|---|---|---|---|---|---|
| 0 | 1.65 | 1.36 | 1.19 | 1.07 | 0.98 | 0.91 | 0.85 | 0.81 | 0.77 |
| 0.2 | 2.32 | 1.82 | 1.54 | 1.37 | 1.24 | 1.15 | 1.07 | 1.01 | 0.96 |
| 0.4 | 4.08 | 2.81 | 2.24 | 1.91 | 1.69 | 1.54 | 1.42 | 1.33 | 1.25 |
| 0.6 | 13.73 | 6.97 | 4.63 | 3.52 | 2.90 | 2.50 | 2.22 | 2.02 | 1.86 |
| 0.8 | 869.00 | 164.50 | 58.54 | 26.30 | 16.04 | 11.05 | 8.27 | 6.56 | 5.43 |

Fuller's (1996) estimator,

$$r = \hat{\rho} + \frac{(1 - \hat{\rho}^2)}{m - 1} \tag{16}$$

which corrects for bias in $\hat{\rho}$, had comparatively low mean squared error and bias over positive $\rho$, but also performed well for negative $\rho$ [22]. The authors particularly recommended Fuller's estimator when sample size is small ($\leq$10) and positive serial correlation is expected [21]. For these reasons, we use $r$ as our serial correlation estimator. We note that for the 2-sample tests, two $\rho$ estimates are computed, one for series A and one for series B, and their average taken after weighting by each series length, $m_A$ and $m_B$.

## 3. Monte Carlo study evaluating statistical properties

### 3.1. Description of Monte Carlo study

We conducted a Monte Carlo study to evaluate Type I error rates, power, and confidence interval width estimation for the serial *t*-tests. For comparison, we also evaluated the same for the usual *t*-tests. Because researchers tend to collect a small number of observations from one individual in an *N*-of-1 trial, we primarily considered *m* from the minimum required up to 12; however, we also included *m* values of 30, 50, and 100 to evaluate large sample properties. We chose serial correlation values of $\rho \in \{-0.33, 0, 0.33, 0.67\}$. The variance remained constant across treatments and series at $\sigma^2 = 1$. For paired *t*-tests, we chose the correlation in pairs to be $\rho_{pair} \in \{0.33, 0.67\}$; 2-sample *t*-tests had $\rho_{pair} = 0$. For level-change tests, the mean structure was E(*Y*) = 1; for rate-change tests, the mean structure was E(*Y*) = $\mu + \beta x$, where $\mu = 0$ and $\beta = 1$. For each combination of the four test types, *m*, $\rho$, and $\rho_{pair}$, we simulated data for 10,000 individuals to maintain Monte Carlo error within 0.01 when estimating proportions with 95% confidence.

### 3.2. Results

For level-change, the paired serial *t*-test has its usual analogue in the 1-sample or Student's *t*-test, and the commonly known 2-sample *t*-test is analogous to the 2-sample serial *t*-test. For rate-change, the paired serial *t*-test has its usual analogue in the *t*-test of a slope in simple linear regression (equivalently, a test of the Pearson correlation coefficient); and the usual *t*-test for a difference in slopes between two treatments is analogous to the 2-sample serial *t*-test.

*Type I error.* We estimated Type I error rates for one-sided 5% significance level tests. The results were similar within the level-change tests and within the rate-change tests. First, the level-change tests: When there is no serial correlation ($\rho = 0$), Type I errors for both the paired and 2-sample tests are at or within a percentage point of nominal levels. The analogous usual tests are at nominal levels as expected. Across all non-zero correlation values, Type I errors are consistently closer to nominal than those for the usual tests. Further, as *m* increases, Type I errors for the serial tests approach nominal, while those for the usual tests show no improvement or get worse. Type I errors for negative $\rho$ are reasonably close (within a couple of percentage points) to a nominal 5% for the paired and 2-sample tests, respectively (top panels of Fig 3 and S1 Fig). For moderately positive $\rho$, Type I errors become reasonable before *m* = 30 for both paired and 2-sample; for high $\rho$, at some point between *m* = 30 and *m* = 50 for both.

Patterns of results for rate-change serial tests differ from those for level-change, but within the rate-change tests, results for the paired and 2-sample are similar. At the smallest values of *m*, Type I errors from the serial tests are higher than their usual *t* analogues. But by *m* = 7 for high positive $\rho$ and *m* = 9 for moderate positive $\rho$, the serial tests perform better than the usual tests (top panels of Fig 4 and S2 Fig). For negative $\rho$, the serial tests are near nominal level. When there is no serial correlation ($\rho = 0$), the two serial tests reject too often for small *m*

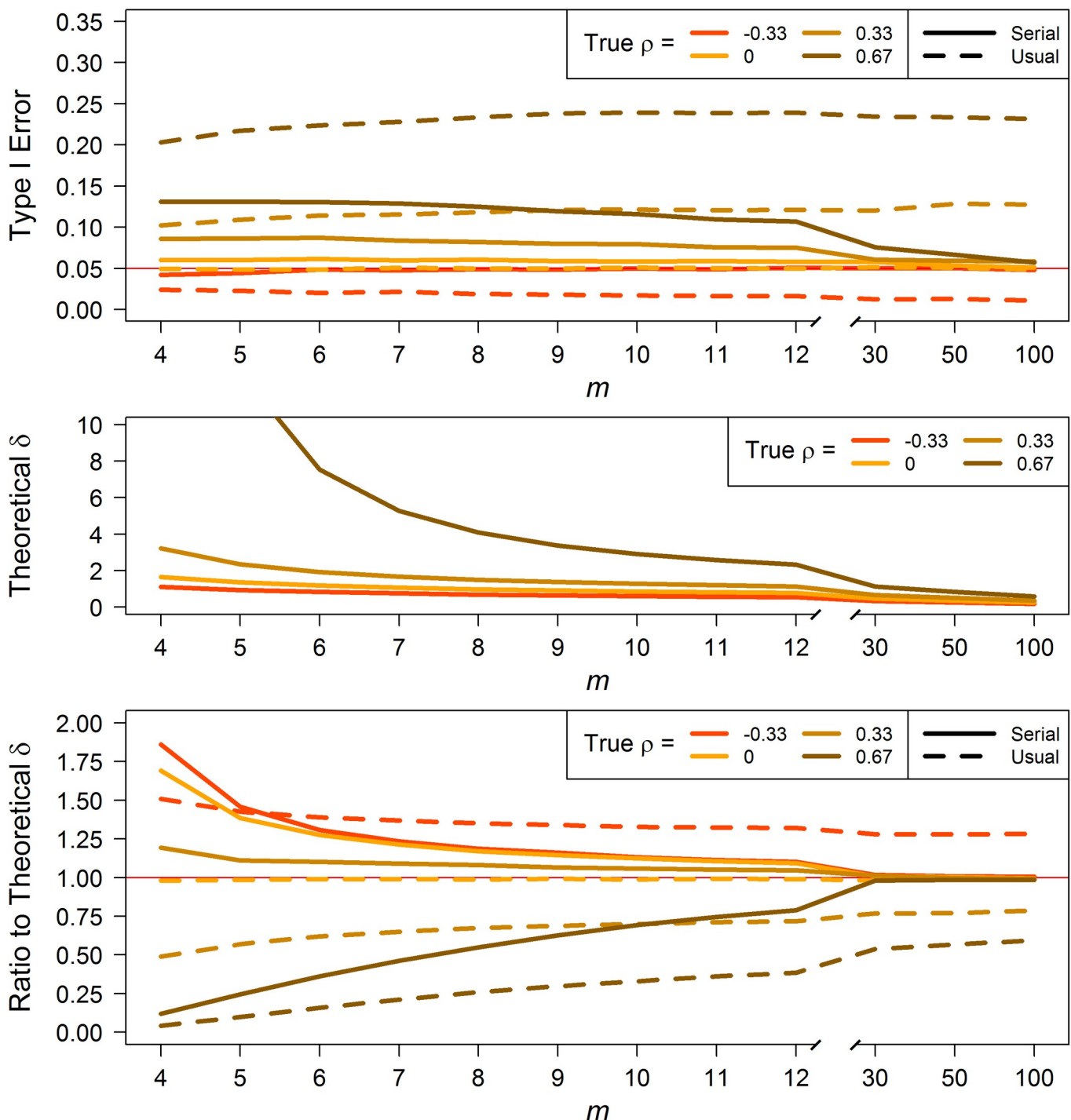

**Fig 3. Paired *t*-test for level-change.** Type I error (top) and theoretical effect size, δ, computed using Eq 8 for 80% power with a one-sided 5% significance level test for a given *m* (middle). Ratios (bottom) of serial and usual δ (estimated from simulation) to the theoretical δ under same conditions as the theoretical δ.

($m{\leq}12$), but approach nominal by $m = 30$. Type I errors for usual *t*-tests (except when $\rho = 0$) are only better than the serial tests for small *m* ($m{\leq}8$), thereafter, usual's Type I errors tend to diverge from nominal level.

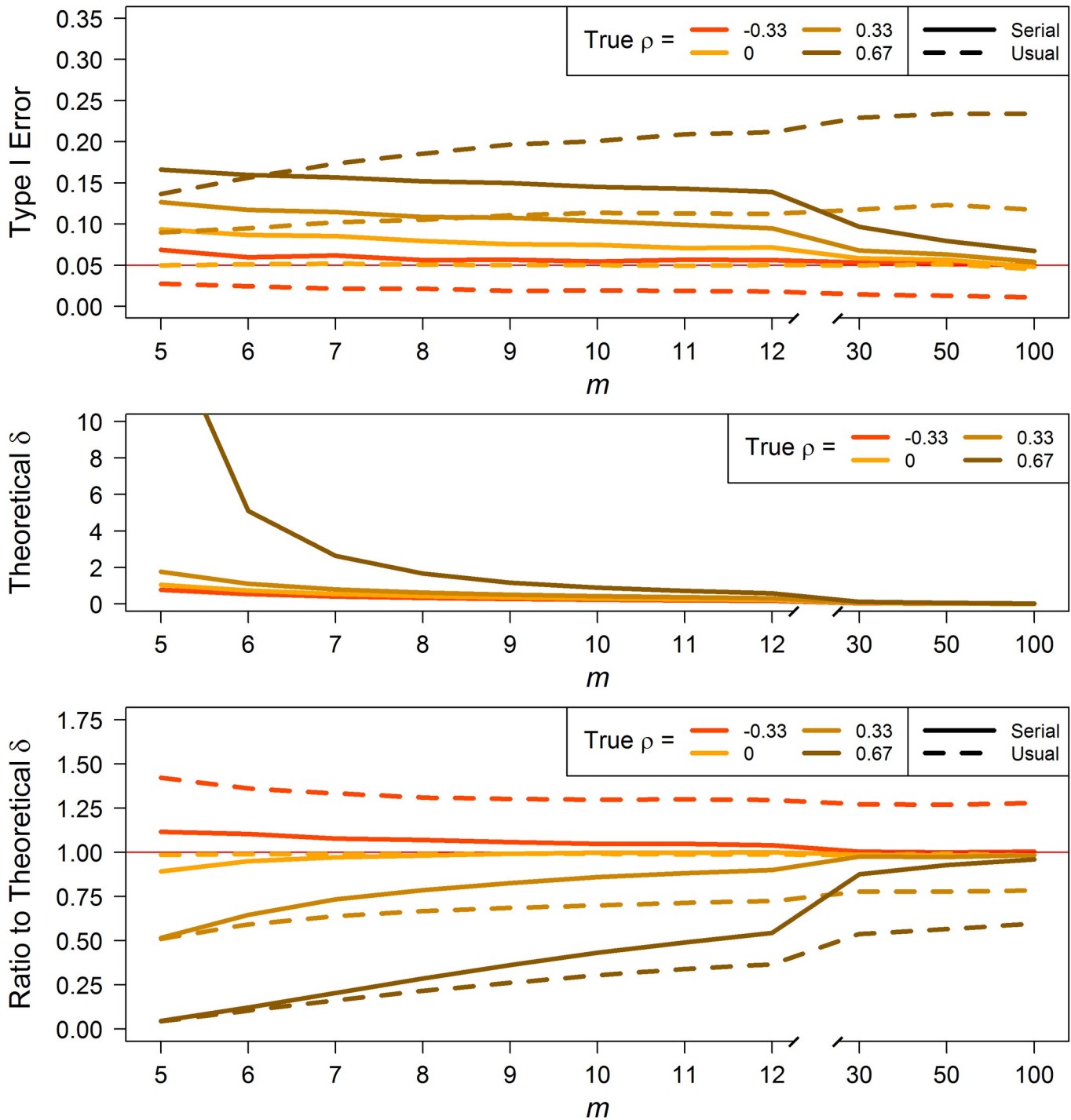

**Fig 4. Paired *t*-test for rate-change.** Type I error (top) and theoretical effect size, δ, computed using Eq 13 for 80% power with a one-sided 5% significance level test for a given *m* (middle). Ratios (bottom) of serial and usual δ (estimated from simulation) to the theoretical δ under same conditions as the theoretical δ.

Regarding moderate vs high $\rho_{pair}$ values in the paired *t*-tests for level- and rate-change, differences in Type I errors are minimal, with the largest difference across 92 relevant Monte Carlo configurations being <1.2 percentage points for serial tests and <1.5 for usual tests.

**Table 3. Factors by which the expected margin of error for a 90% confidence interval (95% one-sided confidence limit) is overestimated or underestimated for select sample sizes (*m*).**

| | Test | Paired test for level change | | | | 2-sample test for level change | | | |
|---|---|---|---|---|---|---|---|---|---|
| $\rho$ | type | *m* = 4 | 8 | 12 | 100 | 2*m* = 8 | 16 | 24 | 200 |
| -0.33 | serial | 2.33 | 1.25 | 1.14 | 1.01 | 1.46 | 1.18 | 1.11 | 1.01 |
| | usual | 1.66 | 1.49 | 1.46 | 1.41 | 1.50 | 1.44 | 1.43 | 1.41 |
| 0.00 | serial | 2.00 | 1.21 | 1.09 | 1.00 | 1.19 | 1.05 | 1.03 | 1.00 |
| | usual | 1.00 | 1.00 | 1.00 | 1.00 | 1.00 | 1.00 | 1.00 | 1.00 |
| 0.33 | serial | 1.22 | 1.11 | 1.04 | 1.00 | 0.80 | 0.89 | 0.92 | 0.99 |
| | usual | 0.43 | 0.59 | 0.64 | 0.70 | 0.57 | 0.65 | 0.67 | 0.71 |
| 0.67 | serial | 0.11 | 0.55 | 0.78 | 0.98 | 0.24 | 0.52 | 0.66 | 0.96 |
| | usual | 0.03 | 0.17 | 0.26 | 0.42 | 0.15 | 0.29 | 0.34 | 0.43 |
| | Test | Paired test for rate change | | | | 2-sample test for rate change | | | |
| $\rho$ | type | *m* = 5 | 8 | 12 | 100 | 2*m* = 10 | 16 | 24 | 200 |
| -0.33 | serial | 1.14 | 1.09 | 1.05 | 1.00 | 1.06 | 1.06 | 1.04 | 1.00 |
| | usual | 1.55 | 1.44 | 1.41 | 1.41 | 1.40 | 1.38 | 1.38 | 1.40 |
| 0.00 | serial | 0.85 | 0.96 | 0.97 | 0.99 | 0.83 | 0.91 | 0.94 | 0.99 |
| | usual | 1.00 | 1.00 | 1.00 | 1.00 | 1.00 | 1.00 | 1.00 | 1.00 |
| 0.33 | serial | 0.45 | 0.71 | 0.83 | 0.98 | 0.55 | 0.71 | 0.80 | 0.97 |
| | usual | 0.47 | 0.60 | 0.65 | 0.70 | 0.62 | 0.67 | 0.69 | 0.71 |
| 0.67 | serial | 0.04 | 0.23 | 0.45 | 0.94 | 0.17 | 0.37 | 0.51 | 0.93 |
| | usual | 0.04 | 0.16 | 0.26 | 0.43 | 0.18 | 0.31 | 0.36 | 0.43 |

## Power

For all four test types, due to the inflated Type I errors for positive $\rho$, power estimates–and therefore effect size estimates–are likely optimistic for both serial and usual tests. The inverse is true for negative correlations. Using the serial *t*-test formulas (Eqs 8, 9, 13 and 15), we can calculate theoretical effect sizes ($\delta$, expressed in terms of $\sigma$) required to achieve a desired amount of power. The middle panels of Figs 3 and 4, and S1 and S2 Figs display the $\delta$ detectable with 80% power with a one-sided 5% significance level test for the given *m* and $\rho$ value. With the simulated data, we estimated the actual $\delta$ detectable under the same assumptions using the serial *t*-tests; we likewise estimated the actual $\delta$ detectable under the same assumptions using the usual *t*-tests. Then we calculated the ratio of actual to theoretical $\delta$ for the serial and usual tests (bottom panels of Figs 3 and 4, and S1 and S2 Figs). This ratio indicates how realistic the serial and usual tests are, with a ratio of 1 being most realistic; i.e., at theoretical $\delta$. For all four test types, across all values of non-zero correlation, the ratios of serial to theoretical $\delta$ are closer to 1 than the ratios of usual to theoretical by *m* = 6, and approach unity as *m* increases, while those for the usual test do not. When there is no correlation ($\rho$ = 0), the usual: theoretical ratio is at 1, as it should be. The discrepancy in the serial:theoretical ratio stems from estimating the "0" correlation, but these ratios also approach 1 with increasing *m*.

## Confidence intervals

We estimated the expected margins of error for 90% confidence intervals for the serial *t* and their usual *t* analogues, and compared these to the mean of $s \times t_{DF, \alpha/2} \sqrt{c(\rho)/b(\rho)}$; see Table 3. For all but the smallest *m*, when $\rho \neq 0$, the margins of error from the serial *t* were closer to theoretical than those based on the usual *t*. Further, as *m* increased, the serial *t*'s margins of error converged to the theoretical; whereas the usual *t*'s margins of error will not converge.

## 4. Examples

### 4.1. Determining benefit of amitriptyline in fibromyalgia

Jaeschke *et al.* (1991) reported results from 23 *N*-of-1 randomized control trials of amitriptyline vs placebo in fibromyalgia patients [23]. Each patient entered an open trial of amitriptyline lasting between 3 and 12 weeks. If there was perceived benefit from amitriptyline, the patient began a double-blind multiple crossover trial. The order of treatment with amitriptyline or placebo was randomized within each of three to six pairs. A pair of treatments lasted four weeks, with two weeks for each treatment. At the end of each week, the patient completed a 7-item questionnaire evaluating severity of symptoms, with each item on a 7-point scale. The two resulting scores within the two-week period were averaged, and the difference between the two treatments' scores within the pair served as the primary outcome. Table 4 contains a portion of the original data from Jaeschke *et al.* The authors individually applied a one-sided (usual) paired *t*-test to determine benefit of amitriptyline for each patient. We re-analyzed each patient's data with the paired serial *t*-test for level change and report one-sided *p*-values in Table 4. However, since this serial *t*-test requires at least 4 pairs of treatments, we could use only 6 of the original 23 patients.

Among the 23 patients (17 not reported here because they had only 3 observations), the pattern of final decisions on amitriptyline continuance was "continue" if the one-sided *p*-value in favor of amitriptyline was ≤ 0.15, and discontinue otherwise [23]. Patients 9, 23, and 17 all exhibited positive serial correlation, which means their SDs were likely underestimated and effective sample sizes overestimated. Had the paired serial *t*-test (for level-change) been used instead of the usual paired *t*-test, decisions for Patients 9 and 23 would have been reversed, and Patient 17 would have been on the cusp. The remaining three patients (18, 15, and 12) exhibited negative serial correlation, and the serial *t*-test corroborated the decisions made using the usual *t*-test.

Zucker *et al.* (1997) re-analyzed these same data to illustrate a hierarchical Bayesian model for *N*-of-1 trials that combined results from all 23 patients to make both population and individual inferences [24]. Their aggregate *N*-of-1 method produced individual posterior probabilities of amitriptyline benefit, say $p(A|y)$, with values closer to 1 indicating benefit. Among the 23 patients, the physician recommended 13 to continue. The minimum $p(A|y)$ in these 13 was 0.85. All 6 patients in our Table 4 had $p(A|y) \geq 0.85$; see Table 1 in [24]. The Bayesian model returned high posterior probabilities of amitriptyline benefit for Patient 23 with $p(A|y) = 0.93$, and Patient 17 with $p(A|y) = 0.94$. These two patients exhibited positive serial correlation; however, the Bayesian model did not account for serial correlation. Had the Bayesian model

**Table 4.** *N*-of-1 randomized control trial data from [23].

| Patient | Consecutive pairs of mean "active vs placebo" differences in 7-item questionnaire | Mean† | SD | Serial *r* | Usual *P* | Serial *P* |
|---------|------------------------------------------------------------------------------------|-------|------|------|-------|-------|
| 9  | 0.05/-0.22/0.57/0.36      | 0.19 | 0.35 | 0.24  | ‡0.10 | 0.25  |
| 18 | 0.64/1.08/-0.36/0.79/-0.64/1.50 | 0.50 | 0.83 | -0.49 | 0.10  | 0.02  |
| 23 | 1.22/1.07/-0.08/0.50      | 0.68 | 0.59 | 0.38  | ‡0.06 | 0.17  |
| 17 | -0.08/0.86/1.07/1.15      | 0.75 | 0.57 | 0.41  | 0.04  | 0.15  |
| 15 | 0.86/1.43/0.65/1.86       | 1.20 | 0.55 | -0.42 | 0.01  | <0.01 |
| 12 | 4.29/3.15/0.78/4.49       | 3.18 | 1.70 | -0.07 | 0.02  | 0.01  |

† Positive means indicate an improvement while taking amitriptyline. The table values in columns 1, 2, and 6 were reproduced from Table 1 on pg 449 of [23].

‡ Based on the data, the usual paired *t*-test returns one-sided *p*-values of 0.18 for Patient 9 and 0.05 for Patient 23.

**Table 5. Example of indifference points from discounting tasks completed pre- and post-treatment for opioid dependence by Patient 1390.**

| Hypothetical delay | 1 day | 1 wk | 2 wks | 1 mth | 6 mth | 1 yr | 5 yr | 25 yr | Level-change | | Rate-change | |
|---|---|---|---|---|---|---|---|---|---|---|---|---|
| | | | | | | | | | *s* | *r* | *s* | *r* |
| Pre-treatment | 92 | 76 | 68 | 58 | 50 | 38 | 18 | 2 | 34.9 | 0.69 | 12.4 | 0.46 |
| Post-treatment | 98 | 92 | 90 | 84 | 72 | 56 | 2 | 2 | | | | |
| Difference | -6 | -16 | -22 | -26 | -22 | -18 | 16 | 0 | 14.2 | 0.50 | 13.7 | 0.32 |

accounted for serial correlation, we suspect the $p(A|y)$ would not have been as high for these two patients, and possibly with $p(A|y) < 0.85$. In summary, the serial *t*-test gave a more conservative inference on amitriptyline benefit than the original (usual) paired *t*-test and subsequent Bayesian model.

## 4.2. Change in delay discounting after treatment for opioid dependence

Landes *et al*. (2012) evaluated each of 159 patients for change in delay discounting–a measure of impulsivity–between pre- and post-treatment for opioid dependence [25]. Just prior to starting, and at the end of a 12-week treatment regimen, patients completed a delay discounting task. The task presented eight series of choices: one series for each of eight hypothetical delays. In each series, the patient was offered choices between a hypothetical $1,000 available after the specified delay or an adjusting amount of hypothetical money available immediately. Each series continued until an indifference point was determined for the specified delay. An indifference point is the percent of the delayed amount the patient deems equivalent to an immediately available amount. Table 5 contains one patient's indifference points at the indicated hypothetical delays in a discounting task completed before (pre) and after (post) a 12-week treatment for opioid dependence. The differences between pre- and post-treatment also are provided. The ordinary least squares standard deviation, *s*, and Fuller's serial correlation estimate, *r*, are based on the indicated model in this paper: level- or rate-change.

To analyze each patient's pre- and post-treatment series of indifference points, Landes *et al*. (2012) used a regression model particular to delay discounting data, and described more fully in Landes *et al*. (2010) [25, 26]. We note their regression model assumed all indifference points were mutually independent. For each patient, they tested, at the 0.05 level, whether discounting post-treatment differed from discounting pre-treatment. We re-analyze their data here.

Their *N*-of-1 design matches a bi-phasic design; hence, we apply the 2-sample serial *t*-tests. Since it is unknown whether patients may have experienced level- or rate-change, we tested for both at the $\alpha/2 = 0.025$ level (i.e., a Bonferroni-corrected 0.05 level). If neither test was

**Table 6. Number (percent) of 119† patients discounting differently between pre- and post-treatment.**

| *N*-of-1 test | Number significant (%) | Median *r* (25%, 75%) |
|---|---|---|
| Original regression method | 69 (58) | - - - |
| *2-sample serial combined* | *22 (18)* | - - - |
| 2-sample serial *t*-tests for level-change | 8 (7) | 0.61 (0.44, 0.69) |
| 2-sample serial *t*-tests for rate-change | 16 (13) | 0.22 (0.02, 0.34) |
| *Paired serial t-tests combined* | *37 (31)* | - - - |
| Paired serial *t*-tests for level-change | 21 (18) | 0.34 (0.01, 0.56) |
| Paired serial *t*-tests for rate-change | 19 (16) | 0.04 (-0.19, 0.32) |

†Out of 159 participants, 3 had two or more sets of data for a single assessment, and 37 exhibited no variability in their indifference points in at least one of the assessments; thus, we excluded these 40 patients.

significant, we interpreted the patient's discounting as having no change; otherwise, the patient's discounting changed between conditions. Table 6 shows the number of patients discounting differently between pre- and post-treatment as determined by the original regression method and by the serial *t*-tests [25]. Table 6 also contains the three quartiles of *r* under each of the serial *t*-test.

Using the 2-sample serial *t*-tests, we found post-treatment discounting to differ from pre-treatment in 22 patients (Table 6). Unsurprisingly, this is less than the 69 patients who were found to have changed discounting using the original regression method, where everything was assumed to be independent [25, 26]. In a study yet to be published, we examined Type I error rates of the regression method and found Type I error rates for nominal 5% tests to be about 15% when true serial correlations were 0.30. In this yet published work, we also recommended pairing responses from two discounting tasks by delay after finding empirical evidence that correlations of paired delays among tasks were, on average, between 0.25 ($N = 394$) and 0.52 ($N = 483$). Therefore, we re-analyzed the original data with the paired serial *t*-tests, testing for both level- and rate-change at the $\alpha/2$ level as before. This time, we found that 37 patients changed discounting (Table 6). Again, this is notably less than the 69 participants who were found to have changed discounting using the original regression method. The decrease was likely due to the Type I errors being closer to nominal with the paired serial *t*-tests. Also, because we account for the correlation between pre- and post-treatment, we find more patients changed using the paired serial *t*-tests than we found using the 2-sample serial *t*-tests, which assume that pre- and post-treatment discounting are uncorrelated. This is because the paired serial *t*-tests have greater power than the 2-sample serial *t*-tests when pairs formed from the two samples are truly (positively) correlated.

Returning to Table 5, the original regression indicated Patient 1390's post-treatment discounting had significantly decreased from pre-treatment levels ($t_{14} = -2.58$, $p = .022$). Treating the pre- and post-treatment discounting datasets as mutually independent, the 2-sample serial *t*-tests for level- and rate-change did not find sufficient evidence this patient had changed (level-change $t_{2.29} = -0.27$, $p = .808$; rate-change $t_{3.98} = -0.61$, $p = .573$). Pairing post-treatment with pre-treatment indifference points, and using the paired serial *t*-tests on the differences also failed to find a statistical change in this patient (level-change $t_{2.22} = -1.32$, $p = .307$; rate-change $t_{2.94} = 0.91$, $p = .432$). For each serial *t*-test considered, the estimated serial correlations ranged from moderate (0.32) to strong (0.69). The original assumption of independence in these data likely led to underestimation of relevant variability and inflation of the effective sample size.

All software code and data used in this paper, are available at github.com in the rdlandes/red-face repository; see files starting with "N-of-1". Alternatively, RDL will provide these files upon request.

## 5. Discussion

Current methods that account for serial correlation in *N*-of-1 data are computationally intensive and often require significant statistical expertise to implement. The serial *t*-tests developed in this paper can accommodate researchers' preferences for simpler methods while still accounting for serial correlation. Type I errors for the serial *t*-tests are closer to nominal level than those for the usual *t*-tests and attain nominal at large *m*. Power for the serial *t*-tests is more realistic (i.e., closer to power computed using the proposed *t*-statistics) than that for the usual *t*-tests; the serial tests also attain theoretical power at large *m*. Margins of error based on the serial *t*-statistics converge to expected width as *m* increases; usual *t*'s margins of error were generally wider when $\rho \neq 0$, and will not attain the expected width no matter the size of *m*.

These serial *t*-tests can be easily implemented by those having only a first course in applied statistics, as they are formulas with no computationally intensive methods needed.

*Limitations*. While the serial *t*-tests demonstrate better Type I error rates, power, and confidence interval width estimation than the usual *t*-tests often used in *N*-of-1 trials, Type I error is still substantial, power optimistic, and interval widths biased for small *m*. This is mainly due to the inaccuracy that remains in estimating $\rho$. Although *r* is bias-corrected, bias still exists; the bias is towards 0 for level-change tests, and is negative for rate-change tests. Type I errors are affected by the biased *r* through the standard errors and degrees of freedom, both functions of $\rho$. The effect of this bias on Type I errors comes more through the estimated degrees of freedom than through the standard errors. However, increasing *m* improves inflated Type I error, optimistic power, and biased margins of error for serial *t*-tests, particularly for the level-change tests; these 3 properties do not improve in the usual *t*-tests.

In the rate-change tests, the estimation of $\rho$ has a strong negative bias. Type I error inflation for these tests is more than that in level-change tests, even with a large number of observations. Further work is needed to correct for the bias in the $\rho$ estimator before these rate-change tests may be applied in clinical use.

Although the serial *t*-tests do not account for carryover effects, the absence of carryover effects is often assumed in applications of *N*-of-1 trials [5]; nevertheless, users need to carefully design experiments to limit any carryover effect that may arise when comparing treatments. These serial *t*-statistics assume that observations are equally spaced in time, with no missing observations. For unequally spaced observations, the estimator of $\rho$ will be biased toward 0. The variance is also assumed to be homogeneous, which is common for most *t*-test applications. For *N*-of-1 trials, this assumption is likely not unrealistic; see Table 1 of Rochon (1990) for an example [8].

As noted above, the $\rho$ estimator still has some bias for small *m*. For a study where multiple *N*-of-1 trials are planned, this bias may be reduced as the *N*-of-1 trials are completed. The $\rho$ estimate for a newly completed *N*-of-1 trial would be a pooled estimate from the completed *N*-of-1 trials, including the newly completed one. Using such a pooled estimator necessarily assumes that $\rho$ is the same for all *N*-of-1 trials (i.e., participants), and that all trials have the exact same temporal spacing of observations (but not necessarily the same *m*). In the end, less bias in the $\rho$ estimates means less bias in the estimated degrees of freedom and estimated standard errors, and more accurate statistical inferences.

The 2-sample serial *t*-tests assume independence between conditions A and B; however, since the data come from the same person, this assumption likely is not true, as illustrated when pairing observations from separate discounting tasks by delay (Section 4.2). When possible, for *N*-of-1 trials with treatments occurring one after the other (as in bi-phasic, pre-post, and ABAB designs), planning the same series for both treatments will allow the use of paired serial *t*-tests, which can have greater power when data between the two treatments are truly pair-wise correlated. The delay discounting example illustrated this characteristic

*Implications*. Accounting for serial correlation makes a difference in statistical inferences and estimation precision for a single individual. Specifically, when serial correlation is positive, finding a "statistical difference" is less likely than when using a method that accounts for serial correlation (e.g., serial *t*-tests). Both examples illustrated this attribute. In the fibromyalgia example, of the six patients showing improvements with methods not accounting for serial correlation (usual paired *t*-tests and a Bayesian aggregate *N*-of-1 model), two patients no longer had a significant result when using a serial paired *t*-test. In the delay discounting example, of the 69 patients originally found to significantly change the way they discounted using a regression method that ignored serial correlation, only 37 patients continued to show evidence of a change when using the paired serial *t*-tests for level- and rate-change. These tempered

statistical results make sense because positive serial correlation reduces the effective sample size, *m*' (Eqs 7 and 12). And, though maybe not as obviously, positive serial correlation increases the standard error of the mean by dividing the usual part of the standard error by a bias that is less than 1; e.g., see the denominator of Eq 4.

As reported by recent reviews of *N*-of-1 trials, the median number of crossovers or repeated cycles is 3; the number of observations, *m*, also tend to be small [3, 13]. From our experience reading various reports of *N*-of-1 trials, *m* = 3 is a typical number in clinical *N*-of-1 trial having multiple crossovers. However, at least 3 data points are needed for estimating parameters $\beta$, $\sigma^2$, and $\rho$, leaving none for error degrees of freedom. That is, *m* should be at least 4 in level-change tests and at least 5 in rate-change tests. Users of these tests should be aware of these limitations and carefully consider the size of *m*.

These serial *t*-tests provide an easy and familiar way to compute effect sizes for a given *m* that will attain a desired amount of power; *a priori* precision calculations are also straight-forward. Such computations give rigor to planning a single *N*-of-1 trial or study. We note, of the very few *N*-of-1 studies we found that provided information on sample size calculations, the "sample size" considered only the number of *N*-of-1 trials that were needed, rather than *m*, the number of observations in a single *N*-of-1 trial. We found no studies using *N*-of-1 trials that considered precision of estimates for an individual trial in their study planning.

For many types of treatments, individual responses are known to differ. Some patients respond well to certain treatments while others show little benefit. This is known as heterogeneity of treatment effects. *N*-of-1 trials evaluate treatment effects on an individual basis; thus eliminating the need to account for heterogeneous treatment effects in analyses. In addition, some patients may be quite variable in their responses, whereas others respond more predictably. The reasons for these heterogeneous treatment effects and variances are not well understood. Therefore, it is important to have an easy-to-use statistical method to identify an appropriate treatment for a particular patient and to also give a more accurate (less-biased) estimate of variance (see Eq 3). Better evaluation of these patient-specific parameters may also help in understanding the mechanism behind these varying effects.

These serial *t*-tests are an improvement over often-employed usual *t*-tests and other methods that fail to account for serial correlation. Additionally, the serial *t*-tests are easy for researchers to implement. Further work is still needed to adjust for *N*-of-1 trials with few observations, particularly for rate-change tests. Nevertheless, we believe these serial *t*-tests will make appropriate analyses for *N*-of-1 trials more accessible to researchers, and allow them to make better decisions for individuals undergoing *N*-of-1 trials.

## Supporting information

**S1 Fig. 2-sample *t*-test for level-change.** Type I error (top) and theoretical effect size, $\delta$, computed using Eq 9 for 80% power with a one-sided 5% significance level test for a given *m*; note, $m_A = m_B$ (middle). Ratios (bottom) of serial and usual $\delta$ (estimated from simulation) to the theoretical $\delta$ under same conditions as the theoretical $\delta$.
(TIF)

**S2 Fig. 2-sample *t*-test for rate-change.** Type I error (top) and theoretical effect size, $\delta$, computed using Eq 15 for 80% power with a one-sided 5% significance level test for a given *m*; note, $m_A = m_B$ (middle). Ratios (bottom) of serial and usual $\delta$ (estimated from simulation) to the theoretical $\delta$ under same conditions as the theoretical $\delta$.
(TIF)

## Acknowledgments

We particularly thank Dr Anne M. Holbrook with McMaster University for her insight into *N*-of-1 trials from a clinical researcher's perspective.

## Author Contributions

**Conceptualization:** Jillian Tang, Reid D. Landes.

**Data curation:** Jillian Tang.

**Formal analysis:** Jillian Tang, Reid D. Landes.

**Investigation:** Jillian Tang.

**Methodology:** Jillian Tang.

**Software:** Jillian Tang.

**Supervision:** Reid D. Landes.

**Validation:** Reid D. Landes.

**Visualization:** Jillian Tang.

**Writing – original draft:** Jillian Tang, Reid D. Landes.

**Writing – review & editing:** Reid D. Landes.

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
