## [Decision Letter · Decision Letter 0]

14 Nov 2019

PONE-D-19-28132

Some t-tests for N-of-1 trials with serial correlation

PLOS ONE

Dear Prof. Landes,

Thank you for submitting your manuscript to PLOS ONE. After careful consideration, we feel that it has merit but does not fully meet PLOS ONE’s publication criteria as it currently stands. Therefore, we invite you to submit a revised version of the manuscript that addresses the points raised during the review process.

We would appreciate receiving your revised manuscript by Dec 29 2019 11:59PM. To enhance the reproducibility of your results, we recommend that if applicable you deposit your laboratory protocols in protocols.io, where a protocol can be assigned its own identifier (DOI) such that it can be cited independently in the future. For instructions see: http://journals.plos.org/plosone/s/submission-guidelines#loc-laboratory-protocols

We look forward to receiving your revised manuscript.

Kind regards,

Alan D Hutson

Academic Editor

PLOS ONE

Journal Requirements:

2. We noted in your submission details that a portion of your manuscript may have been presented or published elsewhere. [Table 4 reproduces a portion of Table 1 from Jaeschke et al. (1991). We re-analyze their data with the proposed statistical methods.]

Please clarify whether this  publication was peer-reviewed and formally published. If this work was previously peer-reviewed and published, in the cover letter please provide the reason that this work does not constitute dual publication and should be included in the current manuscript.

Additional Editor Comments (if provided):

Even though the manuscript is well-written and statistically sound it is unclear what the overall impact of this approach would be in terms of real-world applications. If you could provide an additional real-world example that would satisfy review #1 concerns that would be appreciated.

Reviewers' comments:

Reviewer's Responses to Questions

**Comments to the Author**

1. Is the manuscript technically sound, and do the data support the conclusions?

Reviewer #1: Partly

Reviewer #2: Yes

2. Has the statistical analysis been performed appropriately and rigorously? 

Reviewer #1: Yes

Reviewer #2: Yes

3. Have the authors made all data underlying the findings in their manuscript fully available?

Reviewer #1: Yes

Reviewer #2: Yes

4. Is the manuscript presented in an intelligible fashion and written in standard English?

Reviewer #1: Yes

Reviewer #2: Yes

5. Review Comments to the Author

Reviewer #1:

<h2>**Major points**</h2>

In my opinion, practical aspects of what you are proposing should be discussed more fully.

1) For example, as far as I can see, the methods you develop are for single case n-of-1 trials with many treatment periods per subject. Yet the practical example of fibromyalgia that you use to illustrate your approach fulfills neither of these conditions: many subjects are involved with and most if not all subject have short series for the type of approach your are developing. Of course, one advantage of developing methods for single subjects is that they can be used from the beginning. Methods for many subjects may have to wait a long time for subjects to accrue during which time early patients may be awaiting a decision.

2) Nevertheless, even if you specifically wish to avoid using information from other patients when making a decision for a given one, I think at the very least you ought to consider whether information about nuisance parameters (for example correlation coefficients) at least could be shared between patients.

3)If the design in pairs is used and treatment order is randomised, I would have thought that the naive standard errors for a t-test would be calculated approximately correctly. This is generally what is assumed in experiments subject to temporal or spatial (as in agriculture) correlation. Are you able to check this? Of course, the serial correlation may be more of a problem for a short single series than for the case of multiple sequences considered in the paper by Senn that you cited.

<h2>**Minor points**</h2>

Articles to go with nouns are missing in many places. Some examples are

1) L281 The variance

2)L361 The order

3) L496 The variance

Reviewer #2: Thank you for this contribution.

I propose a few minor changes that should improve the robustness of this manuscript:

Abstract: You do well to highlight the need for this type of work (indeed within patient correlation has been a concern), but the statement "existing methods require simulation..." contradicts your analysis, which itself involves MC simulations. I would clarify your intentions here either in the manuscript or consider revising this portion of the abstract.

Intro: Broaden your review of how many observations within patients are normal- I know some of my collaborators have seen upwards of 30. Review Blackston et al. and Nikles' work for how aggregated nof1s perform regardless of sample size and consider citing.

Methods: Good work overall. Please review Huber et al's work in 2007 on amitriptyline in an Nof1 setting and cite if necessary.

Thanks.

6. PLOS authors have the option to publish the peer review history of their article (what does this mean?). If published, this will include your full peer review and any attached files.

Reviewer #1: Yes: Stephen Senn

Reviewer #2: No

---

## [Author Response · Author response to Decision Letter 0]

19 Dec 2019

We have uploaded a document with our response to reviewers, and a file of R code that addresses a concern from Reviewer #1.

---

## [Decision Letter · Decision Letter 1]

8 Jan 2020

Some t-tests for N-of-1 trials with serial correlation

PONE-D-19-28132R1

Dear Dr. Landes,

We are pleased to inform you that your manuscript has been judged scientifically suitable for publication and will be formally accepted for publication once it complies with all outstanding technical requirements.

With kind regards,

Alan D Hutson

Academic Editor

PLOS ONE

Additional Editor Comments (optional):

Reviewers' comments:

Reviewer's Responses to Questions

**Comments to the Author**

1. If the authors have adequately addressed your comments raised in a previous round of review and you feel that this manuscript is now acceptable for publication, you may indicate that here to bypass the “Comments to the Author” section, enter your conflict of interest statement in the “Confidential to Editor” section, and submit your "Accept" recommendation.

Reviewer #2: All comments have been addressed

2. Is the manuscript technically sound, and do the data support the conclusions?

Reviewer #2: Yes

3. Has the statistical analysis been performed appropriately and rigorously? 

Reviewer #2: Yes

4. Have the authors made all data underlying the findings in their manuscript fully available?

Reviewer #2: (No Response)

5. Is the manuscript presented in an intelligible fashion and written in standard English?

Reviewer #2: Yes

6. Review Comments to the Author

Reviewer #2: (No Response)

7. PLOS authors have the option to publish the peer review history of their article (what does this mean?). If published, this will include your full peer review and any attached files.

Reviewer #2: No

---

## [Editor Report · Acceptance letter]

24 Jan 2020

PONE-D-19-28132R1 

Some t-tests for N-of-1 trials with serial correlation 

Dear Dr. Landes:

I am pleased to inform you that your manuscript has been deemed suitable for publication in PLOS ONE. Congratulations! Your manuscript is now with our production department. 

With kind regards,

on behalf of

Dr. Alan D Hutson 

Academic Editor

PLOS ONE